# Can Fetuin-A, CRP, and WBC Levels Be Predictive Values in the Diagnosis of Acute Appendicitis in Children with Abdominal Pain?

**DOI:** 10.3390/healthcare7040110

**Published:** 2019-09-23

**Authors:** Cengiz Güney, Abuzer Coskun

**Affiliations:** 1Department of Pediatric Surgery, Cumhuriyet University Medical Faculty, Sivas 58140, Turkey; cguney@cumhuriyet.edu.tr; 2Department of Emergency, Sivas Numune Hospital, Sivas 58030, Turkey

**Keywords:** pediatric acute appendicitis, emergency department, perforation, Fetuin-A level

## Abstract

*Background:* Acute appendicitis (AA) is the most common cause of emergency surgery. Therefore, perforation is common. Early diagnosis and new markers are needed. The aim of this study was to investigate the effects of plasma Fetuin-A (FA) levels in patients with an acute abdomen (AB). *Material and Method:* This prospective study included 107 patients younger than 16 years of age who were admitted to the emergency department for abdominal pain between January and December 2018. The patients who presented abdominal pain were divided into two groups as AA and other causes (OC) of AB. Patients with acute appendicitis; intraperitoneal, retrocolic/retrocecal, and appendicitis were divided into three groups. Additionally, the AA group was divided into two groups as perforated appendicitis and non-perforated appendicitis. Serum FA levels of the patients were evaluated in the emergency department. *Results:* In the AA group, C-reactive protein (CRP) and white blood cell (WBC) levels were higher, and FA levels were significantly lower than in the AB group. Intraperitoneal localization was 95.2% and perforation was frequent. When significant values in the univariate regression analysis for acute abdomen and perforation were compared in the multivariate regression analysis, CRP, WBC, and FA levels were found to be prognostic. Furthermore, decreased FA levels were associated with AA, while too greatly decreased FA levels were associated with the risk of perforation. *Conclusion:* Current diagnosis can be made by history, physical examination, laboratory, and imaging methods in appendicitis cases. While trying to diagnose AA in children, the FA, CRP, and WBC levels may be predictive values to identify risk factors.

## 1. Introduction

Appendicitis, the most common cause of surgical abdominal pain in children, is inflammation of appendix vermiformis [1,2]. In 95% of cases, appendicitis is intraperitoneal (in 65%, dorsal to the cecum; in 30%, pelvis), while in 5%, it is retrocolic and retrocecal [3].

The most significant cause of appendicitis in children is lumen obstruction due to an increase in lymphoid tissue. The second most common age range is 10–12. Appendicitis is more common in boys. The possibility of perforated appendicitis in children is higher than in adults. The reason for this is non-specific symptoms, examination findings, and laboratory abnormalities which are thought to be due to lack of communication skills [4].

Laboratory investigations may be performed in patients admitted to emergency services for abdominal pain and suspected appendicitis. However, these results may not be sufficient for diagnosis [3]. White blood cell (WBC) and C-reactive protein (CRP) levels may increase due to various causes such as infection and inflammation. Therefore, it is not reliable in the diagnosis of acute abdomen. High WBC and CRP levels in a case with appendicitis may suggest perforation.

A cheap, reliable, easy, and rapidly-available biochemical marker with high specificity and sensitivity is not yet available in the diagnosis of acute appendicitis (AA). Fetuin-A (FA) is one of the molecules investigated in various fields. Fetuin-A is an insulin-dependent endogenous tyrosine kinase receptor inhibitor, which is mainly synthesized in the liver [5]. Furthermore, it has been found that it directly affects the cells of the animal and human adipose tissue, causing subclinical inflammation and cytokine release [6]. It may be extrahepatically synthesized in the kidneys, choroid plexus, and all vital organs during fetal development. Fetuin-A is seen in the α2 band of serum electrophoresis. The serum concentration is in the range 140–297 mg/L [7]. Fetuin-A was first noted as a negative acute-phase reactant similar to albumin in cases of acute inflammation. Factors affecting FA secretion in humans have been reported such as severe liver damage, cirrhosis, acute viral hepatitis, and cancer [8].

Many studies have examined serum FA levels. However, to the best of our knowledge, no study has been conducted to investigate pediatric acute appendicitis. Since physical examination and laboratory data are insufficient to provide sufficient results, new searches are needed. Therefore, in this study, we aimed to reveal the contribution of FA, WBC, and CRP levels in children to the diagnosis of AA.

## 2. Materials and Methods

### 2.1. Study Design and Population

This prospective cross-sectional study included 107 patients younger than 16 years of age who were admitted to our hospital due to abdominal pain between January and December 2018. The exclusion criteria were as follows: Having known heart and heart valve diseases, drug use due to cardiac causes, having metabolic diseases, chronic liver diseases, chronic renal failure, receiving dialysis treatment, having known inflammatory bowel diseases, malignancies, having known hematological diseases, and receiving erythrocyte suspension over the past six months. All these exclusion criteria were excluded because they could affect acute phase reactants. All cases with suspected acute abdomen except the above exclusion criteria were included in the study. This study was approved by the Local Ethics Committee of the Cumhuriyet University School of Medicine (Decision no: 2019-05/19).

The patients who were admitted to the emergency department with abdominal pain were divided into two groups according to clinical and physical examination and laboratory and imaging results: acute abdomen (AB) and other causes (OC) (urinary tract infections, acute gastroenteritis, renal colic, constipation, etc.). The diagnosis of the acute abdomen was made jointly by emergency medicine specialist, pediatric surgeon, and radiologist according to the results of the clinical examination, physical examination findings, laboratory results, and radiological imaging results. Surgery is still the gold standard in the diagnosis of acute appendicitis. Non-surgical cases were diagnosed with physical examination, laboratory, ultrasonography, contrast-enhanced abdominal tomography or magnetic resonance imaging. The patients diagnosed with acute abdomen were divided into three groups: Those without appendicitis and those with intraperitoneal appendicitis and retrocolic/retrocecal appendicitis. Acute appendicitis was divided into two groups as perforated and not perforated.

To determine serum Fetuin-A levels, 5 mL of venous blood was collected from the patients presenting with abdominal pain. The blood was centrifuged at 4000 rpm for 5 min. Serums were kept in Eppendorf tubes and kept at −80 °C. Fetuin-A levels were analyzed by Human FETUA (Fetuin-A) Sandwich Enzyme-Linked Immunosorbent Assay (ELISA) kit (96-Fine Test, EH0218, Wuhan Fine Biotechnology, China). Range was 140–297 mg/L and sensitivity was <0.469 ng/mL.

Hemogram was measured using a Beckman Coulter Automated CBC Analyzer (Beckman Coulter, Inc., Fullerton, CA, USA).

Biochemistry blood was analyzed with the Cobas 6000 (C6000-Core, Cobas c-501 series, Hitachi, Roche, Indianapolis, IN, USA).

### 2.2. Statistical Analysis

The data obtained from this study were analyzed by SPSS 15.0 (SPSS Inc., Chicago, IL, USA) software package. While determining the normality of the variables, Shapiro–Wilks was used because of the number of units. When analyzing the differences between the groups, Mann–Whitney U Tests were used because the variables did not show normal distribution. The chi-square analysis was performed to examine the relationships between the groups of nominal variables. In the cases where the expected values in the 2 × 2 tables did not have sufficient volume, Fisher tabs exact test was used, and in the R × C tables, Spearman correlation analysis was performed with the help of Monte Carlo simulation. Besides, linear regression analysis was used for univariate and multivariate analysis of variables. We used univariate analysis to measure the relationship of variables in patient and control groups. The variables that were found to be statistically significant in univariate analysis were used in a multivariate linear regression risk model with a forward step method to determine the independent prognostic factor. When interpreting the results, the significance level was set at 0.05 and *p*-values less than 0.05 were considered as statistically significant. Demir study was used in the preparation of collecting the data [9].

### 2.3. Study Limitations

The most significant limitation of the study was that it was single-centered and had a low number of cases. Additionally, working with children was another critical limitation. The refusal to participate in the study by some patients, the cost of the FA kit, and the lack of adequate financial support were other challenges of the study.

## 3. Results

The clinical and demographic characteristics of the patients are listed in Table 1. According to the Mann–Whitney U test, significant differences were found between the groups in terms of age, sex, and CRP, WBC, and FA levels. CRP and WBC levels were significantly higher while the FA level was considerably lower in the AA group than in the OC group. Mean corpuscular volume (MCV) and amylase were also statistically significant.

Chi-square analysis of AA and OC groups revealed a significant difference between the groups in terms of sex, location of AA, radiological image, and perforation (Table 2).

Chi-square analysis of the location of AA revealed a significant difference between the groups in terms of sex, presence of AA, radiological image, and perforation (Table 3).

Chi-square analysis of perforated appendicitis revealed a significant difference between the groups in terms of sex, the location of AA, radiological image, and other causes of abdominal pain (Table 4).

The univariate regression analysis of groups of abdominal pain revealed no significant differences in terms of alanine aminotransferase (ALT), alkaline phosphatase (ALP), and amylase, while significant differences in terms of CRP, WBC, FA, AA, perforation, age, sex, aspartate aminotransferase (AST), and radiological imaging were found. However, after the adjustment of the variables that are statistically significant in univariate analysis in the multivariate linear regression analysis with advanced stage method, FA, CRP, and WBC levels increased and remained associated with the risk of acute abdomen (Table 5).

Univariate regression analysis with perforated appendicitis revealed no significant difference in terms of sex, while significant differences in terms of CRP, WBC, FA, age, AST, ALT, ALP, amylase, and radiological imaging were found. However, after the adjustment of the variables that are statistically significant in univariate analysis in the multivariate linear regression analysis with advanced stage method, FA, CRP, and WBC levels increased and remained associated with the risk of perforation (Table 6).

There was a statistically significant difference between the acute abdomen and perforated appendicitis in terms of age, FA, CRP, WBC, and sex. Fetuin-A had a strong negative correlation, while the other variables had strong positive correlation.

## 4. Discussion

The key to successful treatment in acute appendicitis is early and accurate diagnosis [10]. However, the correct diagnosis rate is 72%–94%. Negative appendectomy rates vary between 15%–34%. These rates are proof that diagnosis is still difficult [11,12,13]. We aimed to determine the relationship between serum FA levels and disease activity and inflammatory parameters for early diagnosis and prognosis in pediatric acute appendicitis patients. Our study is the first to report that decreased FA levels are independent predictors of disease activity in patients with AA and that serum FA levels are negatively correlated with CRP concentrations and WBC count. This suggests that Fetuin-A may have a possible inflammatory function in AA and can potentially be used as a biological marker.

The most common cause of obstruction in the child appendix lumen is lymphoid tissue hyperplasia. There is a positive correlation between the severity of the inflammatory event in the appendix and the likelihood that the lumen will become obstructed. Due to the secretion of the appendix mucosa, fluid accumulation and distension develop rapidly in the cavity. The appendix mucosa may continue secretion even when the pressure in the lumen is high. Due to these reasons, the appendix first gets gangrene, and then it gets perforated. Besides, the proliferation of bacteria living in the appendix due to the closed space contributes to this process [3,14,15,16].

In our study, we detected appendicitis in 46 (42.9%) patients. Of these patients, 39 (36.4%) had intraperitoneal and seven (6.5%) had retrocecal and retrocolic appendicitis. These patients underwent a standard appendectomy. Appendiceal perforation was observed in 26 (37.7%) patients. These patients underwent standard appendectomy followed by drainage. Fecalith was detected in 11 (18.3%) of the patients with perforation and only two (4.5%) of the patients without perforation.

Fetuin-A is a glycoprotein [17,18,19], also known as 2-Heremans Schmid. Its molecular weight is about 60 kDa [19]. It consists of a long-chain A and a short B chain connected by a short peptide. Before FA, synthesized in a single chain, becomes mature with its two-chain form in circulation, it undergoes posttranslational modification processes such as proteolysis, glycosylation, and phosphorylation. It is high in serum (140–297 mg/L) [18,19,20]. Fetuin-A is a negative acute phase reactant [17,18,19]. Fetuin-A levels were found to be low in acute alcoholic hepatitis, acute drug-associated hepatitis, chronic autoimmune hepatitis, fatty liver patients, alcoholic, and primary cerebrospinal cirrhosis and hepatocellular cancer patients [20]. Serum fetuin-A levels were found to be low in patients with end-stage renal disease who commonly develop cardiovascular calcification [21]. Low fetuin-A levels have been shown to be associated with an increased risk of death in dialysis patients, independent of diabetes and inflammation [22]. Manolakis et al. [23] demonstrated that the decrease in FA showed a close association with the acute phase and that chronic inflammation in both Crohn’s and ulcerative colitis might be a potential diagnostic and perhaps predictive value molecule. In another study, it was reported that serum levels decrease in response to infection and/or inflammation, play a role as an anti-inflammatory mediator, and have protective effects against lipopolysaccharide-associated shock [24].

In our study, we found a significant decrease in serum FA levels in AA patients compared to the OC group. The mean serum FA level was 273 mg/dL in the OC group and 176 mg/dL in the AA group. Fetuin-A value was 223 mg/dL in the group without perforation and 161 mg/dL in the group with perforation. These values were significant and predictive. This suggests that FA may also play a role in the pathophysiology of AA as a negative inflammatory mediator. In our study, we found a significant relationship between serum FA levels in the AA group and perforated appendicitis group. This relationship was as meaningful as the WBC count and CRP levels. There was a strong negative correlation between serum FA levels and CRP and WBC levels in the acute appendicitis group. FA was found to be significant in both univariate and multivariate regression analysis of both AA and perforation. In the perforated appendicitis group, where inflammation was more frequent, serum FA levels were found to be lower than the non-perforated AA group. These results indicate that serum FA level can be used as an essential marker in the pathology of appendicitis. Therefore, its clinical significance should be interpreted with caution.

Increased levels of CRP determine the presence and severity of inflammation. Wang et al. [25] found an inverse relationship between CRP levels and Fetuin-A and reported that this inverse relationship was present between FA and inflammation. Ketteler et al. [26] reported that the low FA level in patients with chronic renal failure who underwent stable hemodialysis was inversely related to CRP, an indicator of inflammation.

The negative relationship between serum FA levels and CRP levels in our study was consistent with the literature. Additionally, we demonstrated that serum FA levels in patients with AA correlate negatively and strongly with CRP concentrations and WBC count. While all of these values (Table 6) were significant in univariate regression analysis of perforated appendicitis group, only WBC, CRP, and FA levels were significant in multiple linear regression analysis. Therefore, acute phase reactants were high in AA patients with high inflammation, while the negative phase reactant FA was low. Acute phase reactant is a sensitive marker of CRP and tissue damage and systemic inflammation. The degree of inflammation in the acute appendicitis was low with high WBC and CRP levels, and it correlated with serum FA concentration. Serum FA levels were significantly lower in patients with AA compared to perforated appendicitis. In pediatric abdominal pains, low serum FA levels, and high CRP and WBC levels, physical examination, and radiological imaging increase the accuracy rate in the diagnosis of acute appendicitis.

As a result, the decrease in serum FA levels was associated with the disease in AA and perforation patients. Serum FA levels were negatively correlated with C-reactive protein levels and WBC count. Serum FA levels were found to be lower in patients with perforated appendicitis than AA patients. In the perforation prediction, CRP and WBC levels were very high, and FA levels very low was guided. Fetuin-A may be an important indicator for perforation to drop too. Further research may examine the use of Fetuin-A concentration measurements as markers of disease activity in OC patients.

## 5. Conclusions

In acute appendicitis and perforated appendicitis groups, negative acute phase reactant was found to have a significant relationship with serum FA level and other inflammatory parameters. This finding is consistent with other studies that detected serum FA as a negative acute phase reactant. Measurement of serum FA levels in pediatric patients with abdominal pain can be used as a test for the diagnosis of appendicitis. Low serum FA levels; will increase the rate of correct diagnosis, while reducing the rate of negative appendectomy. Prospective randomized trials to be conducted with more patient groups are needed on this issue.

## Figures and Tables

**Table 1 healthcare-07-00110-t001:** Baseline characteristics of study patients.

Abdominal Pain					
	All Patients	Patients with		*Z*	*p*-Value
		OAP Group	AA Group		
**Baseline Characteristics**					
**Age, mean ± SD, year**	9.04 ± 2.60	8.05 ± 2.12	10.35 ± 2.61	−4.524	0.001
**Sex, Female/Male**	46/61	33/28	13/33	x^2^ = 7.143	0.008
**Laboratory Finding**					
**AST**, mg/dL	28.37 ± 16.04	25.10 ± 10.57	32.71 ± 20.57	−1.274	0.202
**ALT**, mg/dL	27.48 ± 18.26	24.52 ± 16.26	31.42 ± 20.14	−1.921	0.056
**ALP**, mg/dL	96.63 ± 40.63	92.70 ± 9.23	100.69 ± 45.03	−1.366	0.176
**CRP**, mg/L	4.19 ± 3.42	2.11 ± 3.38	6.95 ± 4.14	−7.039	0.001
**Amilaz**, U/L	90.63 ± 39.13	84.24 ± 38.24	99.11 ± 39.08	−2.380	0.001
**WBC**, 10^3^/uL	13.12 ± 4.69	9.93 ± 2.75	17.36 ± 3.10	−8.282	0.001
**MCV**, fL	87.35 ± 9.61	85.60 ± 7.42	89.67 ± 6.67	−2.189	0.029
**MCH**, pg	29.31 ± 3.01	29.15 ± 3.19	29.53 ± 2.78	−0.758	0.448
**MCHC**, g/dL	33.35 ± 5.63	33.69 ± 0.79	32.89 ± 0.85	−0.265	0.791
**RDW**, %	14.54 ± 1.52	14.17 ± 1.60	14.64 ± 1.41	−1.266	0.206
**MPV**, fL	8.37 ± 1.14	8.27 ± 1.02	8.50 ± 1.29	−0.753	0.453
**Fetuin-A**, mg/L	231.86 ± 50.54	273.43 ± 10.60	176.72 ± 20.42	−8.829	0.001

OAP: Other Abdominal Pain; AA: Acute Appendicitis; AST: Aspartate Aminotransferase; ALT: Alanine Aminotransferase; ALP: Alkaline Phosphatase; CRP: C Reactive Protein; WBC: White Blood Cell; MCV: Mean Corpuscular Volume; MCH: Mean Corpuscular Hemoglobin; MCHC: Mean Corpuscular Hemoglobin Concentration; RDW: Red Cell Distribution Width; MPV: Mean Platelet Volume; * *p* < 0.05.

**Table 2 healthcare-07-00110-t002:** Chi-Square test results relating to the difference between variables of abdominal pain.

Abdominal Pain					
		OAP Group	AA Group	*X* ^2^	*p*-Value
		*n* (%)	*n* (%)		
**Gender**	**Female**	33 (30.8)	13 (12.2)	7.143	0.008
	**Male**	28 (26.2)	33 (30.8)		
**Radiological Imaging**	**USG**	24 (22.4)	33 (30.8)	11.056	0.001
	**Abdominal CT**	37 (34.6)	13 (12.1)		
**Acute Abdomen**	**No**	61 (57.0)	0 (0)	107.00	0.001
	**Intraperitoneal**	0 (0)	39 (36.4)		
	**Retrocolic/retrocecal**	0 (0)	7 (6.5)		
**Ferforation**	**NO**	61 (57.0)	20 (13.1)	24.39	0.001
	**Yes**	0 (0)	26 (37.7)		

USG: Ultrasonography; CT: Computed Tomography; * *p* < 0.05.

**Table 3 healthcare-07-00110-t003:** Chi-Square test results relating to the difference between variables of acute abdomen.

Acute Abdomen						
		No	IP	RC/Rc	*X* ^2^	*p*-Value
		*n* (%)	*n* (%)	*n* (%)		
**Gender**	**Female**	33 (30.8)	8 (7.5)	5 (4.8)	13.421	0.008
	**Male**	28 (26.2)	31 (29.0)	2 (1.8)		
**Radiological Imaging**	**USG**	24 (22.4)	27 (25.2)	6 (5.6)	11.703	0.001
	**Abdominal CT**	37 (34.6)	12 (11.2)	1 (0.9)		
**Abdominal Pain**	**OAP**	61 (57.0)	0 (0)	0 (0)	107.00	0.001
	**AA**	0 (0)	39 (36.4)	7 (5.6)		
**Perforation**	**No**	61 (57.0)	20 (13.1)	0 (0)	50.030	0.001
	**Yes**	0 (0)	19 (17.8)	7 (5.6)		

IP: Intraperitoneal; RC/Rc: Retrocolic/Retrocecal; * *p* < 0.05.

**Table 4 healthcare-07-00110-t004:** Chi-Square test results relating to the difference between variables of perforation.

Perforation					
		OAP Group	AA Group	*X* ^2^	*p*-Value
		*n* (%)	*n* (%)		
**Gender**	**Female**	39 (36.5)	7 (6.5)	3.618	0.070
	**Male**	42 (39.2)	19 (17.8)		
**Radiological Imaging**	**USG**	36 (33.6)	21 (19.6)	11.056	0.001
	**Abdominal CT**	45 (42.1)	5 (4.7)		
**Acute Abdomen**	**No**	61 (57.0)	0 (0)	107.00	0.001
	**Intraperitoneal**	0 (0)	19 (17.8)		
	**Retrocolic/retrocecal**	0 (0)	7 (6.5)		
**Abdominal Pain**	**OAP**	61 (57.0)	0 (0)	24.39	0.001
	**Yes**	20 (18.7)	26 (24.3)		

**Table 5 healthcare-07-00110-t005:** Univariate and multivariate linear regression analyses for predicting the development of acute abdomen.

Acute Abdomen										
	Univariate				Multivariate					
	RS	F	β	t	*p*-Value	RS	F	β	t	*p*-Value
**CRP**	0.297	44.316	0.545	6.657	0.001			0.428	3.741	0.041
**WBC**	0.619	170.816	0.787	13.07	0.001	0.952	153.921	0.542	4.814	0.013
**Fetuin-A**	0.906	1009.443	−0.952	−31.772	0.001			−0.848	−11.124	0.001
**AA**	0.854	615.920	0.924	24.818	0.001			0.265	3.972	0.004
**Perforation**	0.426	77.818	0.652	8.821	0.001			−0.253	−6.239	0.001
**Age**	0.194	25.203	0.440	5.020	0.001					
**Gender**	0.067	7.511	0.258	2.741	0.007					
**AST**	0.056	6.198	0.236	2.489	0.014					
**ALT**	0.035	3.839	0.188	1.959	0.053					
**ALP**	0.010	1.014	0.098	1.007	0.316					
**Amilaz**	0.036	3.887	0.189	1.971	0.051					
**RI**	0.103	12.009	−0.321	−3.478	0.001					

RS: R Square; RI: Radiological Imaging; F: Anova test; β: Beta; t: student test, * *p* < 0.05.

**Table 6 healthcare-07-00110-t006:** Univariate and multivariate linear regression analyses for predicting the development of perforation.

Perforation										
	Univariate				Multivariate					
	RS	F	β	t	*p*-Value	RS	F	β	t	*p*-Value
**CRP**	0.377	63.512	0.614	7.969	0.001			0.312	4.034	0.037
**WBC**	0.482	97.668	0.694	9.883	0.001	0.779	27.547	0.298	5.531	0.023
**Fetuin-A**	0.630	178.971	−0.794	−13.378	0.001			−1.476	−7.539	0.001
**AA**	0.505	107.047	0.711	10.346	0.001			0.330	2.160	0.001
**AB**	0.426	77.818	0.652	8.821	0.001			−1.157	−6.239	0.031
**Age**	0.064	7.179	0.253	2.679	0.009					
**Gender**	0.034	3.674	0.184	1.917	0.058					
**AST**	0.883	23.581	0.428	4.856	0.001					
**ALT**	0.154	19.132	0.393	4.374	0.001					
**ALP**	0.061	6.789	0.246	2.606	0.011					
**Amilaz**	0.069	7.818	0.263	2.796	0.006					
**RI**	0.098	11.344	−0.312	−3.368	0.001					

RS: R Square; RI: Radiological Imaging; AB: Acute Abdomen; * *p* < 0.05.

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
