# Peer review of "Can Fetuin-A, CRP, and WBC Levels Be Predictive Values in the Diagnosis of Acute Appendicitis in Children with Abdominal Pain?"

_healthcare, 2019, doi:10.3390/healthcare7040110_

Round 1

Reviewer 1 Report

Good

Author Response

First of all, thank you.

1- Language revision was performed.

2-The research was adapted to the design.

3-The results are presented more clearly.

Reviewer 2 Report

In this single-center prospective study, the authors report the usefulness of measuring serum Fetuin-A levels in the diagnosis of acute appendicitis in children, but there are many concerns about this report. First, inclusion and exclusion criteria in this study should be clearly described. In addition, the gold standard for the diagnosis of appendicitis seems to be surgery, but it is questionable how to diagnose cases without surgery. As a result of this study, Fetuin-A shows a negative correlation with WBC and CRP. Inmultivariate analysis of appendicitis diagnosis and perforation prediction, not only Fetuin-A but also WBC and CRP are significant factors as predictors. A major problem is that the clinical significance of Fetuin-A added to WBC and CRP in the diagnosis of appendicitis is unclear.

Author Response

First of all, thank you.

1- Language was performed.

2- The research was adapted to the design.

3- The result is presented more clearly.

4- The results were further supported by the findings.

   Also;

    Inclusion and exclusion criteria were not clearly defined. How non-surgical cases were diagnosed was defined. The clinical uncertainty associated with fetuin-a was corrected.

Reviewer 3 Report

Thanks for letting me review this paper. I enjoyed reading the paper. Please see my comments below -

Abstract

1) Line 13 – Please review “Perforation is more common than adults”. I think this is an incomplete sentence. Are you suggesting that perforation in children is more common than adults?? If yes, please make appropriate revisions.

2) Line 19 – I think there is an extra T (please check). I would suggest revising the sentence. Patients with acute appendicitis were divided into three groups:…..

3) Line 25 – 27 – I would encourage authors to revise these statements (for clarity purpose).

Introduction

1) Line 55 - I would suggest rephrasing the sentence “To the best of our” – Many studies….however, there exists scarcity of research…

2) Towards the end of introduction section, please add 2-3 sentences that provide overview of the paper. This will remind readers of the purpose and significance of the project.

Material and Methods

1) Line 62 - Authors have provided a list of exclusion criteria. Is there a reason behind the big list of exclusion criteria? If yes, please provide a brief description of the rationale behind this.

2) I did not see any supporting evidence/cited literature in Statistical Analysis section. I would suggest adding literature/citations to support your points.

Discussion

This section is well written, however, references are not current (not published within last 5 years). I reviewed the reference list and seems like authors included only 1 manuscript/article from 2017. I would encourage authors to find current research in the field and make appropriate revisions. This will also help with the literature review (in introduction section).

Author Response

First of all, thank you.

1- The entrance was reorganized to provide adequate infrastructure and to include all relevant references.

2- Language revision was performed.

3-The research was adapted to the design.

4-The results are presented more clearly.

5- The results were further supported by the findings.

Also;

Line 13 and Line 19 misspelling corrected. Line 25-27 and Line 55 were further strengthened. Towards the end of the introduction, 2-3 sentences have been added that provide an overview. Line 62 exclusion criteria were adjusted. The literature was added to the statistics section.  The sources have been updated as much as possible without compromising the discussion.

Round 2

Reviewer 2 Report

The author well revised the manuscript. I have no more comments.